# Supplementation with *Silybum marianum* Extract, Synbiotics, Omega-3 Fatty Acids, Vitamins, and Minerals: Impact on Biochemical Markers and Fecal Microbiome in Overweight Dogs

**DOI:** 10.3390/ani14040579

**Published:** 2024-02-09

**Authors:** Fatemeh Balouei, Bruno Stefanon, Elisa Martello, David Atuahene, Misa Sandri, Giorgia Meineri

**Affiliations:** 1Department of Agrifood, Environmental and Animal Science, University of Udine, Via delle Scienze 206, 33100 Udine, Italy; balouei.fatemeh@spes.uniud.it (F.B.); misa.sandri@uniud.it (M.S.); 2Division of Epidemiology and Public Health, School of Medicine, University of Nottingham, Nottingham City Hospital Campus, Nottingham NG5 1PB, UK; martello.elisa@gmail.com; 3Department of Veterinary Sciences, School of Agriculture and Veterinary Medicine, University of Turin, Grugliasco, 10095 Turin, Italy; david.atuahene@unito.it (D.A.); giorgia.meineri@unito.it (G.M.)

**Keywords:** overweight, supplement, dogs

## Abstract

**Simple Summary:**

The effects of a 35-day administration of a supplement containing extracts from *Silybum marianum*, prebiotics, probiotics, n-3 polyunsaturated fatty acids, minerals, and vitamins were considered in our research involving 10 overweight adult dogs. The supplement demonstrated positive impacts on ALP, glucose, direct bilirubin, and CRP, with a decrease observed during the 35-day administration period. Fecal microbiota diversity decreased just after 7 days from the beginning of supplement administration and recovered from day 14 onwards. However, there was significant individual variability in the interaction between the supplement and the microbiome. Overall, the study suggests a potential benefit of the supplement in supporting the health of overweight dogs.

**Abstract:**

Overweight and obese dogs can develop metabolic dysfunction, characterized by an inflammatory response and involvement of liver functions. If a modulation of the gut microbiome and its interaction with the gut–liver axis is implicated in the development of metabolic dysfunction, exploration becomes necessary. Over the past decade, diverse therapeutic approaches have emerged to target pathogenic factors involved in metabolic dysfunction. This study investigated the impact of a supplement with hepatoprotective activity, containing extracts of *Silybum marianum*, prebiotics, probiotics, n-3 polyunsaturated fatty acids, vitamins, and minerals on hematological markers of liver functions and inflammation, as well as on the intestinal microbiota of 10 overweight adult dogs over a 35-day time span. Animals underwent clinical and laboratory evaluations every 7 days, both before the administration of the supplement (T0) and after 7, 14, 21, 28, and 35 days (T1, T2, T3, T4, and T5). In comparison to T0, a significant (*p* < 0.05) decrease in ALP, glucose, direct bilirubin, and CRP was observed from T3 to T5. The alpha diversity of the fecal microbiota significantly decreased (*p* < 0.05) only at T1, with high variability observed between dogs. Total short-chain fatty acid and lactic acid were also lower at T1 (*p* < 0.05) compared to the other times of sampling. The beta diversity of the fecal microbiota failed to show a clear pattern in relation to the sampling times. These results of blood parameters in overweight dogs show a reduction of the inflammation and an improvement of metabolic status during the study period, but the effective contribution of the supplement in this clinical outcome deserves further investigation. Furthermore, the considerable individual variability observed in the microbiome hinders the confident detection of supplement effects.

## 1. Introduction

Nutritionists have become increasingly aware of the importance of prevention in maintaining the health of companion animals. This heightened awareness has led to a surge in the popularity of dietary supplements for pets [1], typically consisting of one or more ingredients intended to complement their diet. These supplements serve the purpose of preventing nutritional deficiencies and often contain essential components like vitamins, minerals, amino acids, and essential fatty acids. They can also provide other functional compounds, such as herbs, nutraceutical plants, enzymes, and metabolites. In addition to their preventive role in disease development, nutritional and functional supplements also fall under the realm of complementary or alternative medicine. These supplements provide an alternative or adjunct approach to drug therapies, offering a means to minimize side effects [2]. The hepatoprotective activity of *Silybum marianum* in canine medicine has been reported [3,4,5] in relation to its antioxidant and anti-inflammatory activities. The liver plays a vital role in regulating the body’s internal antioxidant levels, primarily through the synthesis of glutathione, a significant intracellular antioxidant. Recurrent factors in obesity are the disrupted oxidative balance and alteration of liver function [6], along with an inflammatory response [7], emphasizing the importance of a well-rounded diet in achieving a favorable oxidative balance in dogs. Furthermore, Bahmani et al. [8] reported that the activity of *Silybum marianum* extends beyond hepatoprotection; it also has hypolipidemic and cardiovascular properties, preventing insulin resistance.

Oxidative stress arises when the production of reactive oxygen species (ROS) exceeds the body’s capacity to counteract them with antioxidant defense mechanisms. To effectively manage oxidative stress and inflammation, antioxidant supplementation with vitamins E and C is often recommended [9], in addition to nutraceuticals. These vitamins aid in building a robust antioxidant network, replenishing glutathione levels, and restoring the activity of glutathione peroxidase in the livers of dogs [10,11]. However, there are few studies on the effects of vitamins E and C on the treatment of liver diseases [12], and the American College of Veterinary Internal Medicine in 2019 [13] reported that no clinical study relating the effect of these vitamins on liver disease had been published at that time.

Choline and betaine are methyl donors and, together with sulfur-containing amino acids, participate in the synthesis of S-adenosylmethionine and glutathione [14]. In the biosynthesis of S-adenosylmethionine (SAM), choline and betaine act as intermediates and play a vital role. They act as precursors to phosphatidylcholine, enabling the transportation of fat from the liver to muscles and adipose tissue [15]. These compounds play a crucial role in the regulation of antioxidant defenses and in the anti-inflammatory response [16]. Omega-3 fatty acids are other compounds fed to dogs to control the inflammatory process [17]. Supplementation with a mixture of compounds is a strategy aimed at potentiating the control of oxidative stress and inflammation [4,18,19].

Moreover, there is a significant focus on understanding the role of gut microbiota in animal health [20,21] and in obesity [22,23], also considering the relationship between the gut and liver. This connection, known as the gut–liver axis [24], is maintained through a complex interplay of metabolic, immune, neuroendocrine, and microbial interactions, ensuring the regulation and stability of the gut–liver axis relationship via the portal circulation [25]. The tight junctions of the intestinal epithelium act as a natural barrier against bacteria and their metabolic by-products. When antigens, whether from pathogenic microorganisms or food, cross these junctions, they trigger an immune response that alters the behavior of T lymphocytes (T-regulatory cells and T-helper 17 cells), resulting in the release of inflammatory cytokines and chemokines that ultimately enter the portal circulation [26]. 

Given the influence of the microbiome on metabolic dysfunction, it becomes crucial to explore substances that can positively modulate the microbiota, preventing dysbiosis and inflammatory processes at the intestinal mucosal level. 

Probiotics, prebiotics, and synbiotics have gained significant recognition as potential treatment options for liver disease in humans [27]. The concept of “synbiotics” refers to the combination of prebiotics and probiotics, acting synergistically by providing the former with substrates for the latter to better survive within the gut, thus promoting the production of short-chain fatty acids (SCFAs). These SCFAs can serve as energy substances for intestinal cells [21], mimicking the effects of probiotics [20]. In this way, prebiotics contribute to promoting a healthy intestinal environment and fostering a symbiotic relationship between the host and beneficial microorganisms [28,29]. In particular, prebiotics are substances that stimulate the growth or activity of microorganisms, functioning as undigested foods that aid in intestinal peristalsis and selectively promoting the growth of beneficial intestinal bacteria. Certain prebiotics, such as fructo-oligosaccharides (FOS), have demonstrated potential in reducing signs of liver damage in humans by restoring levels of beneficial bacteria, making them a promising therapeutic option [30,31]. 

Considering the positive activity of natural compounds on liver and metabolic functions and gut health, in the present study, we wanted to investigate the safety and efficacy of a supplement made with a mix of nutraceuticals in overweight dogs. We evaluated changes in biochemical parameters and assessed the impact on the composition of the gut microbiome.

## 2. Materials and Methods

### 2.1. Animals and Study Design

A total of 10 adult overweight dogs were carefully selected from a veterinary clinic in Italy (Table 1). The inclusion criteria for dogs in the study were overweight and body condition score (BCS) higher than normal. Enlisted dogs exhibited at least one of the altered liver parameters: AST, ALT, ALP, GGT, or direct bilirubin. Additionally, subjects had at least one altered metabolic parameter: glucose, triglycerides, or CRP higher than the reference value. On the basis of the medical history and general clinical examinations by the veterinarian, other diseases, such as kidney disease, dermatitis, otitis, inflammatory bowel disease, and diabetes were excluded, and dogs were free from internal and external parasites, and not under pharmacological treatments for the last 30 days. The study was approved by the Ethical Committee of the University of Turin (Protocol N° 1234 of 14 April 2022). After obtaining informed consent from the pet owners, the dogs remained domiciled at their usual site for the entire duration of the study, ensuring minimal disruption to their daily routines.

At the commencement of the trial (T0) and subsequently during every week of treatment, comprehensive nutritional assessments and physical examinations were conducted by the clinicians on the subjects to evaluate their overall health and monitor any changes. Body weight (kg) and body condition score (BCS) were recorded at the beginning of the study. BCS assessments were conducted by a trained veterinarian who assigned values ranging from one to nine [32], with a BCS score of four or five considered the ideal value according to FEDIAF guidelines [33]. 

Throughout a period of 42 days, dogs were closely monitored and observed, with regular check-ups conducted at the veterinary facility. To establish the daily portion of the maintenance diet, the estimated requirements were computed following NRC guidelines [34] using the value of 110 kcal of metabolizable energy per kg metabolic live weight. Since the aim was to evaluate the effects of the supplement on blood and fecal parameters, the owners were asked to feed the amount of feed indicated on the commercial label of the complete diets. After 7 days of adjustment of the amount of food, the subjects continued to be fed with the same regular maintenance adult diet. After these 7 days, a supplementary feed, consisting of 1 tablet (2 g) per 15 kg of body weight, was given every 24 h, which continued to be provided throughout the study duration. 

The supplement contained a combination of natural ingredients as reported in Table 2. The formulation of supplements was based on the published data by Marchegiani et al. [35].

At weekly intervals, following a longitudinal study design, the attending veterinarian conducted thorough clinical examinations of the subjects and evaluated biochemical data to monitor the general health of the animal. Fecal and blood samples were collected starting at 7 days from the enrolment of the dog (T0) and then after 7, 14, 21, 28, and 35 days (T1, T2, T3, T4, and T5, respectively).

### 2.2. Laboratory Analysis

#### 2.2.1. Blood Analysis

Blood tests on serum samples were conducted to assess specific parameters linked to liver functions and metabolism, including aspartate transaminase (AST), alanine transaminase (ALT), alkaline phosphatase (ALP), gamma glutamyl transaminase (GGT), glucose, triglycerides, direct bilirubin, and C-reactive protein (CRP). For the biochemical parameters, we used Catalyst I (IDEXX Laboratories, Inc.; Westbrook, ME, USA) and for the CRP parameter, we utilized the rapid test TestICT (Diagnosticavet srl, Milan, Italy).

#### 2.2.2. Microbiome Analysis

Fecal samples were collected by the owner on the day of the enrolment (T0) and before each of the other scheduled weekly appointments. Owners were instructed to collect stools in a plastic sterile collection tube and to store them immediately at −20 °C. The samples were delivered in an ice pack box to the clinic on the day of the appointment and stored at −20 °C. At the end of the 35 days of the trial, all fecal samples were delivered in an ice pack box to the veterinary clinic and immediately sent to the laboratory for analysis [36].

To quantify the content of SCFAs, namely acetate, propionate, butyrate, isobutyrate, isovalerate, and lactic acid in fecal samples, high-performance liquid chromatography (HPLC) equipment was employed (Shimadzu Corporation, Kyoto, Japan). Fecal samples (1 g) were diluted 1:20 with H_2_SO_4_ 0.1 N (Carlo Erba, Milan, Italy) and briefly vortexed, and then centrifuged at 4000× *g* for 15 min in a centrifuge (5804R, Eppendorf AG, Hamburg, Germany). Supernatants were filtered through syringe filters (RC 0.45 μm, 25 mm, DTO Servizi Srl, Venice, Italy) and the filtrate was transferred into an autosampler vial. Twenty microliters (20 μL) were injected into HPLC Shimadzu LC-20AT, equipped with a Prominence SPD-M20A photodiode-array detector. For separation, an Aminex HPX-87H column (300 mm × 7.8 mm) and a pre-column (Bio-Rad, Hercules, CA, USA) were used. Peaks of analytes were identified by comparing the retention times of standard mixtures to those of the samples, and quantification was based on peak area measurements by the external standard method. Standards of acetic, propionic, butyric, isobutyric, isovaleric, valeric, and lactic acid were obtained from Merck (Darmstadt, Germany).

Total DNA was extracted from 150 mg of feces using a fecal DNA miniprep kit (Zymo Research; Irvine, CA, USA), following the manufacturer’s instructions. After the extraction, quantification and quality assessment were performed using a QubitTM 3 Fluorometer (Thermo Scientific; Waltham, MA, USA). Libraries were prepared by amplifying the V3 and V4 regions of the 16S rRNA gene, incorporating indexes for sequencing. This was accomplished using a Nextera DNA Library Prep kit (Illumina; San Diego, CA, USA), following the manufacturer’s instructions and utilizing specific primers [37]. The resulting amplicons were sequenced on a MiSeq platform (Illumina; San Diego, CA, USA) in 2 × 250 paired-end mode, adhering to standard procedures and generating a sequencing depth of 50,000 reads.

The effectiveness of the entire pipeline, from DNA extraction to taxonomic annotation, was assessed using a ZymoBIOMICSTM Microbial Community Standard (Zymo Research, Irvine, CA, USA). According to the manufacturer’s specifications, the mock community included eight bacterial species with their corresponding percentages: *Pseudomonas aeruginosa* (4.2%), *Escherichia coli* (10.1%), *Salmonella enterica* (10.4%), *Lactobacillus fermentum* (18.4%), *Enterococcus faecalis* (9.9%), *Staphylococcus aureus* (15.5%), *Listeria monocytogenes* (14.1%), and *Bacillus subtilis* (17.4%). 

### 2.3. Bioinformatic and Statistical Analysis

The initial raw sequences (FASTQ) from the 10 DNA isolation samples were processed using the bioinformatic tool Quantitative Insights Into Microbial Ecology 2 (QIIME 2) [38]. After demultiplexing, the sequenced reads meeting the quality threshold (Phred score ≥ 30) were identified, and chimeras were filtered out. The remaining high-quality sequences were then clustered into amplicon sequence variants (ASV) against the Greengenes database [39] for 16S rRNA. Alpha diversity between dogs and times of treatment was reported as a Shannon rarefaction curve, and comparison of evenness was also computed. To evaluate beta diversity, the phylogeny was constructed based on the weighted UniFrac distance metric [40], and the results were visualized using Principal Coordinate Analysis (PCoA) plots. The differences in community composition were assessed by performing a permutational multivariate analysis of variance (PERMANOVA) using weighted UniFrac distances.

The normalized relative abundance profiles (RAs) of taxa were obtained for annotated sequences for each sample. Data were uploaded into the MicrobiomeAnalyst suite [41] for further statistical evaluation, including LefSe, single analysis, and multifactorial analysis, with times of sampling as the main factors. In the multifactorial analysis, the dog was considered to be a covariate to determine its impact on community composition. The raw sequence data were subsequently uploaded to the NCBI Sequence Read Archive (PRJNA1051361).

Blood variables were analyzed using the mixed model procedure of SPSS v. 25 [42] with time as a repeated factor and subjects as a random factor. 

## 3. Results

The mean ± sd values of enzymatic activity (AST, ALT, ALP, and GGT) and concentrations of glucose, triglycerides, direct bilirubin, and CRP in the serum sampled from the dogs before and after the administration of a supplement over a time span of 35 days are reported in Table 3. 

The activity of AST at the start of the study, before supplement administration (T0), was 49.5 UI/L and decreased to 31.0 UI/L after 14 days (T2; *p* < 0.05). Subsequently, there was an increase in AST, and the values were not different from T0 or T2. The mean ALT activity was higher than the reference value at T0 and decreased from T0 to T5, with the lowest values at T1, but these differences were not significant. The mean ALP decreased from the initial values (T0) until T3 (*p* < 0.05). Although there was a decrease in ALP, all mean values for all time points remained higher than the maximum reference value. 

The mean GGT activity did not vary between sampling times, although a slight decrease was observed. At T0 and T1, the mean glucose concentrations were significantly (*p* < 0.05) higher than at T3, T4, and T5, but all the values were within the reference range. Triglycerides showed a trend similar to glucose, with a constant decrease from T0 to T5, but due to wide individual variability, the changes in concentration were not significantly different. At time point T0, the direct bilirubin concentration was 0.6 mg/100 mL, which was above the normal reference range. At T1 and T2, the direct bilirubin concentration was reduced to 0.4 mg/100 mL, and fell within the normal reference range for T3, T4, and T5 (*p* < 0.05). The initial CRP concentration (T0) was 1.2 mg/L, exhibiting a significant decrease (*p* < 0.05) until the sampling time T3 (0.6 mg/L), and it maintained stability throughout the remaining duration of the study.

The evaluation of the effect of supplementation on the gut microbiome was considered in terms of the composition of the bacterial population and the main products of fermentation, namely SCFAs and lactic acid.

Figure 1A contains data representing the values of the Shannon index of alpha diversity in the intestinal microbiota of the 10 adult dogs over the course of a complementary diet intervention at different time points (T0, T1, T2, T3, T4, and T5). There was not a large variation in the index for each time of sampling, and the *p*-value of the ANOVA test indicates that there were no statistically significant differences in these values, suggesting limited dynamic changes in the taxa abundances during the dietary intervention. 

The data provided in Figure 1B represent the alpha diversity values for each dog during the study and highlight the wide variations between the subjects (*p* < 0.05). Some dogs exhibit alpha diversity values higher than three (D8, D1, D10), while others have values around two (D6, D2). Conversely, the rarefaction curves clearly indicated that the sequence depth was appropriate to describe the diversity of the microbiome in the fecal samples.

The evenness, calculated starting from the alpha biodiversity, was significantly lower (*p* < 0.05) at T1 compared to T0. For the samples collected after T1 (Figure 2A), the evenness was not significantly different from T0. However, the variability between dogs was significant (Figure 2A), with D6 having the lowest and D8 the highest values. Of note, the within-dog variability of evenness was very reduced for four dogs (D5, D3, D8, and D10) in comparison to the other subjects, confirming the large individual variability.

The beta diversity was assessed using weighted UniFrac distances (Figure 3A) for adult dogs over 35 days (T0 to T5). The graphical appraisal showed that the samples from T0 to T5 did not cluster and were almost scattered, underlining limited variations during the time course of the study. The results of beta diversity distances for different dogs showed two distinct clusters of samples representing dogs D7 and D4, while the samples from other dogs were more scattered and tended to overlap (Figure 3B).

As a final evaluation of the variation of the fecal microbiome in relation to supplement administration, the comparison of relative abundances (RAs) between times of sampling was computed and features significantly different are reported in Figure 4. 

*Ruminococcus gnavus* showed a large fluctuation over the specified time points. Notably, the highest RA of *Ruminococcus gnavus* was at T3 and T5 (58.4 and 58.6, respectively) and these values were higher than those measured at T0 and T1 (25.6 and 23.7, respectively; *p* < 0.05). Apart from the decrease from T3 to T4, we can consider that this species increased after the administration of the supplement. The variations in RA of *Blautia* in adult dogs with liver damage after the administration of the supplement dramatically decreased (*p* < 0.05) from T0 to the other time points of sampling. At T0, the highest RA of *Blautia* accounted for 45.5, indicating a high presence of this microbial species at the beginning of the study, and the lowest was 11.8 at T3. The RAs of *Streptococcus* were very low from T0 to T3 and sharply increased at T4 and T5, reaching values of 9.0 and 5.7 (*p* < 0.05).

In relation to the initial RA at T0, *Turicibacter* showed a fluctuation, decreasing until T3 (*p* < 0.05), followed by an increase to a basal value at T5, and the mean values ranged from 7.1 to 25.1. The RA of the family of *Peptostreptococcaceae* remained relatively stable at T0 and T1, notably increased at T2 (33.5), and had the lowest values at T3 (15.5) and T4 (17.1). These differences were statistically significant (*p* < 0.05).

Table 4 reports the concentrations of total SCFA in μmol/g and lactic acid in percentage at the different time points (T0 to T5) in 10 overweight adult dogs. The concentrations of SCFA and lactic acid and were significantly (*p* < 0.05) lower at T1 (213.9 μmol/g) and significantly (*p* < 0.05) higher at T3 (249.7 μmol/g). Lactic acid concentrations were lower at T1 (0.4 μmol/g) and were significantly (*p* < 0.05) higher at T3 (2.6 μmol/g) and T5 (2.5 μmol/g). 

During supplement administration, lactic acid levels displayed dynamic fluctuations, initially declining and subsequently increasing, hinting at possible shifts in metabolic processes. Meanwhile, the levels of acetic acid, propionic acid, isobutyric acid, butyric acid, and isovaleric acid remained relatively constant throughout the study, showing no significant changes.

## 4. Discussion

The study involved 10 adult dogs fed with a supplement over the course of 35 days, with a focus on assessing the impact on overall well-being and biochemical parameters. Additionally, the study aimed to evaluate changes in the composition of the intestinal microbiota. 

According to the inclusion criteria and biochemical data (Table 3), dogs were overweight, with associated liver damage, and had one or more blood parameters out of range of the reference value. In humans, metabolic syndrome is a complex dysfunction that increases the risk of cardiovascular disease, liver damage, and non-insulin dependent diabetes mellitus [43]. In dogs, metabolic syndrome is not yet clearly recognized [44], and a similar condition is reported as obesity-related metabolic dysfunction. Dogs suffering from metabolic dysfunction associated with obesity show higher ALT, ALP, glucose, and triglycerides in comparison to healthy dogs [45]. Obese dogs often exhibit oxidative stress and inflammation [6,7], which, in turn, can affect liver functions. As reported by Gommeren et al. [46] and Torrente et al. [47], a high concentration of CRP is associated with a systemic inflammatory condition and can be used to monitor the progress of recovery during treatments.

After the administration of the supplement, a significant improvement in ALP, glucose, direct bilirubin, and CRP was observed starting from T3 (Table 3), supporting a positive effect on liver functions and metabolic condition in the 10 overweight dogs.

The supplement contained different compounds, and it is not feasible to understand the main roles of each of them. The use of a multi-compound supplement is a limitation of the study, since this does not allow for the clear elucidation of compound-specific activity. On the other hand, in clinical practice, supplements often consist of a blend of different ingredients that require testing in in vivo trials.

*Silybum marianum* is a biennial plant belonging to the *Asteraceae* family. The predominant active constituent of the extract is silymarin, a phytocomplex composed of various flavonoids, with silybin, isosilybin, silychristin, and silydianin being the most significant ones, primarily concentrated in its fruits and seeds [48]. In humans, silymarin possesses multiple beneficial effects for hepatobiliary disease treatment, encompassing antioxidant, anti-inflammatory, and antifibrotic properties [49,50]. Supplementation with *Silybum marianum* to support the treatment of liver disease could also be postulated in dogs [5,51]. In a previous study of dogs with liver disease [5], the plasma activity of ALT was significantly reduced, and paraxonose activity increased after 60 days of treatment with silymarin. Moreover, the mRNA expression of mitochondrial *SOD_2_* was significantly upregulated. In the study by Gogulsky et al. [52] the oral administration of silymarin in dogs with hepatopathy resulted in a significant decrease in the plasma activity of ALT, AST, and GGT. Other compounds with anti-inflammatory activity are EPA and DHA, which mediate the release of proinflammatory cytokines [53]. Specifically, EPA stimulates the biosynthesis of resolvins, and DHA stimulates the biosynthesis of resolvins, maresins, and protectins. In humans, the dietary administration of n-3 PUFA is beneficial for the treatment of non-alcoholic fatty liver disease [54,55]. In dogs, there is no proven evidence for the efficacy of EPA and DHA on liver disease, but anti-inflammatory activity is reported for other conditions, such as atopic dermatitis [17] and atrial fibrillation [56].

Although ALP, CRP, glucose, and direct bilirubin could indicate a recovery of the dogs from inflammation and metabolic dysfunction, the fecal microbiome showed only a transient decrease in alpha diversity after 7 days (T0) of supplement administration (Figure 2, *p* < 0.05).

The administration of *Bacillus subtilis* to weaned puppies, together with other ingredients, caused only minor modification to the RA of fecal microbiota [57]. The species is considered positive for the gut health of dogs, affecting the production of SCFAs [58,59]. But, in the present study, changes in the microbiome were not in line with previous studies. Our results agree with those of Isidori et al. [60] for dogs with chronic inflammatory enteropathy supplemented with a dose of 125 billion *B. Subtilis*, which is comparable to what we used. The authors did not find significant variations in SCFAs, alpha diversity, and beta diversity in the fecal samples in relation to the administration of *B. Subtilis*.

The supplement fed to the dogs in the present study also contained FOS and beta glucans from barley, which are prebiotics known to modify fecal microbiota and end products of fermentation in dogs [61,62,63]. However, the amounts of prebiotics contained in our supplement was much lower than those used in these studies, and they are likely to have a very small effect on the gut bacterial population. In another study, FOS were administered either independently or in conjunction with mannan-oligosaccharide to dogs whose diet primarily consisted of meat. This resulted in elevated ileal immunoglobulin A concentrations but reduced concentrations of fecal total indole and phenol [64]. Flickinger et al. [65] reported that in adult male beagles, an oligofructose-enriched diet decreased fecal ammonia and *Clostridium perfringens* concentrations, while total aerobes increased, thus ameliorating overall dog health.

Nutraceuticals from plant extracts can also modify gut microbiota, and polyphenols have gained the attention of researchers of dogs [18,36,66]. However, to the best of our knowledge, no results are available in the literature concerning the effect of *Silybum marianum* on the fecal microbiota of dogs, although silybin has been reported to positively affect the microbial population of the gut. Interestingly, studies conducted by Wang et al. [67] indicated a beneficial effect of silybin on the gut microbiota of mice.

The slight variation observed in the fecal microbiome of dogs was probably due to the different effects of each compound contained in the supplement on microbial populations. However, other factors can be implied, as deducted from the large individual variability of the fecal microbiome at the beginning of the study and later after feeding the supplement (Figure 2 and Figure 3).

In our study, the fluctuation in the relative abundance of *Turicibacter*, the linear decrease in *Blautia* with the times of sampling, and the higher relative abundance of *Streptococcus* at T4 and T5 (Figure 4) would suggest that the supplement decreased the overall health condition of the gut. In the dysbiosis index proposed by AlShawaqfeh et al. [68], *Blautia* and *Turicibacter* contribute positively to gut health, while *Streptococcus* has a negative impact. Conversely, the relative abundance of *Ruminococcus gnavus* was higher at the end of the study (T5) in comparison to T0 and T1. This species, or at least some strains, is thought to have beneficial roles in the guts of humans [69] and possibly in dogs [10]. *Blautia*, *Turicibacter*, and *Ruminococcus gnavus* are considered butyrate-producing bacteria [27,70,71,72] and their variation would impact SCFA fecal content. However, the results reported in Table 4 did not show any variation in individual and total SCFA, apart from T1. It is likely that in the complex microbiota ecosystem of the gut, the ecological niche of these bacteria was occupied by other taxa. 

The study has some limitations, including the relatively small sample size of dogs and the absence of a control group. Additionally, the dogs were kept on their usual diets, making it challenging to determine the specific quantities of the nutrients they consumed. Consequently, the interpretation of the study’s findings is confined to understanding the impact of adding a supplement within the context of a typical nutrient intake.

## 5. Conclusions

Our study and results highlight the individualized nature of the diet–microbiota relationship in overweight adult dogs. The dietary supplement had positive effects on blood parameters associated with the liver and on metabolic dysfunction. The dogs displayed considerable inter-individual variation in microbiota composition, and overall, the supplement’s impact on the gut microbiota was complex and not uniform. These findings underscore the importance of considering individual variability when assessing the effects of dietary interventions on canine health. The limited number of studies in the literature emphasizes the need for further research to understand the effectiveness of dietary supplements for metabolic dysfunction and associated liver alterations in overweight dogs. 

## Figures and Tables

**Figure 1 animals-14-00579-f001:**
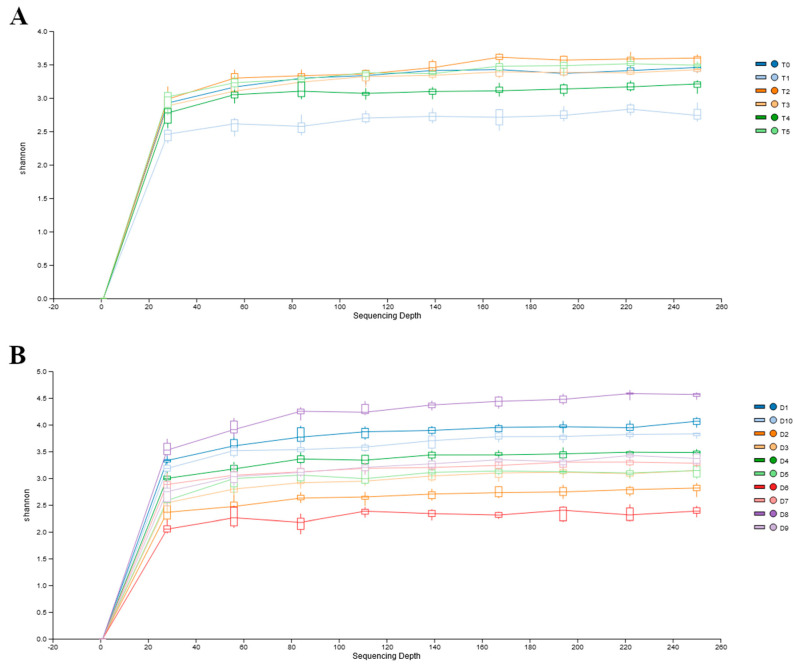
Rarefaction curves of the alpha diversity Shannon index between days of sampling (**A**), and between dogs participating in the study (**B**). Panel (**A**): T0, before administration of supplement; T1, T2, T3, T4, and T5 correspond to 7, 14, 21, 28, and 35 days from the administration of supplement; Panel (**B**): each rarefaction curve corresponds to the six samples for each of the dogs reported in the legend.

**Figure 2 animals-14-00579-f002:**
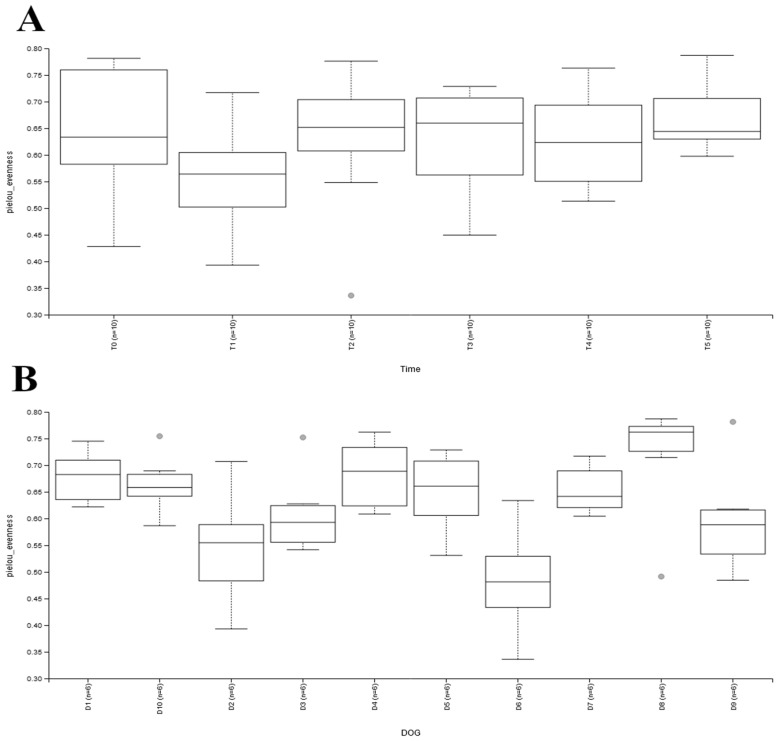
Evenness of the samples between days of sampling (**A**); and between dogs participating in the study (**B**). Panel (**A**): T0, before administration of supplement; T1, T2, T3, T4, and T5 correspond to 7, 14, 21, 28, and 35 days from the administration of supplement; Panel (**B**): the box plots correspond to the six samples for each of the dogs.

**Figure 3 animals-14-00579-f003:**
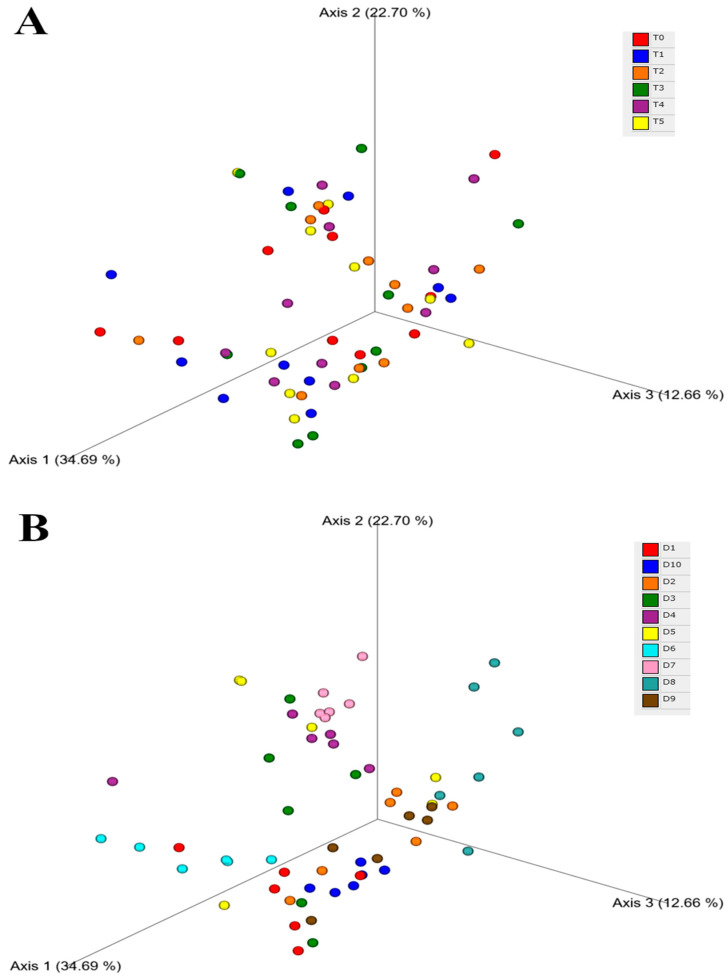
Principal coordinates analysis of weighted UniFrac distance metric for beta diversity between days of sampling (**A**); and between dogs (**B**). Panel (**A**): T0, before administration of supplement; T1, T2, T3, T4, and T5 correspond to 7, 14, 21, 28, and 35 days from the administration of supplement; Panel (**B**): D1 to D10 denotes the dogs.

**Figure 4 animals-14-00579-f004:**
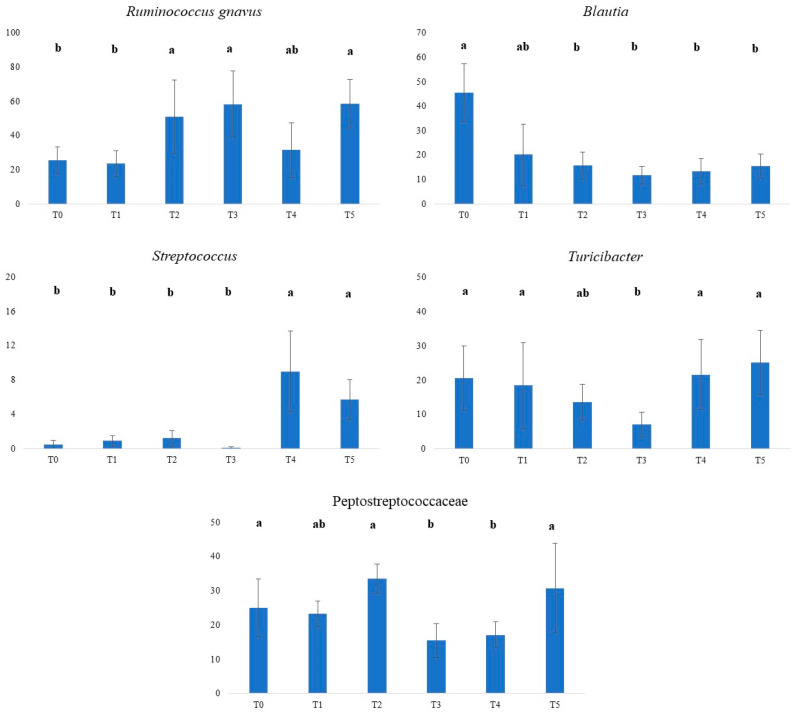
Normalized counts (*y* axis) of *Ruminococcus gnavus*, *Blautia*, *Streptococcus*, *Turicibacter*, and *Peptostreptococcaceae* in 10 adult dogs with liver disease at various time points (*x* axis). T0, before administration of supplement; T1, T2, T3, T4, and T5 correspond to 7, 14, 21, 28, and 35 days from the administration of supplement; a, b denotes a mean that significantly differs for *p* < 0.05.

**Table 1 animals-14-00579-t001:** Data on included dogs. Age (years), sex, breed, weight, and body condition score.

Dog Number	Age	Sex	Breed	Weight (kg)	BCS	Tablet
D1	12	F	American Staffordshire terrier	30	7	2
D2	10	F	Dachshund	9	8	0.75
D3	8	M	English Bulldog	30	8	2
D4	8	M	Shar Pei	35.5	7	2.3
D5	12	M	Staffordshire bull terrier	22.1	7	1.5
D6	11	F	Beagle	17.5	8	1
D7	13	M	Bullterrier	38.1	7	2.3
D8	10	F	Dalmatian	28.8	7	2
D9	9	F	German shepherd	34.8	6	2
D10	8	M	Basset hound	30.5	8	2

Data were recorded at the beginning of the trial (T0); BCS: scores 0–9; sex: M = male; F = female.

**Table 2 animals-14-00579-t002:** Formulation of the supplement administered to the dogs for the study.

Diet	Amount
*Sylibum marianum* extract	2.52%
Choline chloride	0.80%
*Bacillus subtilis*	6.40%
Fructoligosaccharides	0.50%
Beta glucans from barley	0.50%
Betaine	0.90%
n-3 polyunsaturated fatty acids (n-3 PUFA) in fish oil *	6.40%
Vitamin B12	0.00002%
Vitamin C	0.15%
Alpha tocopherol	3.40%
Zinc	0.93%
Magnesium	1.65%
Arginine	0.40%
Taurine	2.40%
Hydrolyzed proteins	24.30%
Maltodextrine	48.75%

* Fish oil contains 27% EPA and 16% DHA.

**Table 3 animals-14-00579-t003:** Plasma enzymatic activity, metabolites concentration, and C-reactive protein concentration in the plasma of dogs with liver disease before and after the administration of the supplement.

Item	RV		T0		T1		T2		T3		T4		T5	
AST	23–66	Mean	49.5	a	32.8	ab	31.0	b	34.3	ab	36.0	ab	40.5	ab
UI/L		sd	21.1		16.8		11.5		14.0		14.5		14.0	
ALT	21–102	Mean	107.2	ns	82.4	ns	91.2	ns	91.6	ns	96.0	ns	91.1	ns
UI/L		sd	45.9		36.4		69.6		62.2		70.2		65.9	
ALP	20–156	Mean	302.1	a	246.0	ab	242.9	ab	208.6	b	213.4	b	204.6	b
UI/L		sd	241.5		220.3		239.2		211.8		202.7		201.7	
GGT, UI/L	1.2–6.0	Mean	6.3	ns	6.2	ns	6.3	ns	6.2	ns	5.4	ns	5.2	ns
UI/L		sd	3.5		2.3		3.8		3.2		2.8		2.8	
Direct Bilirubin	0.1–0.5	Mean	0.6	a	0.4	ab	0.4	ab	0.3	b	0.3	b	0.3	b
mg/100 mL		sd	0.2		0.1		0.2		0.1		0.1		0.1	
Glucose	65–110	Mean	106.4	a	108.6	a	97.4	ab	92.9	b	93.9	b	91.2	b
mg/100 mL		sd	15.2		30.0		24.8		14.8		22.8		13.7	
Tryglicerides	20–112	Mean	162.7	ns	119.1	ns	150.6	ns	110.5	ns	114.6	ns	103.1	ns
mg/100 mL		sd	94.6		61.0		103.7		54.5		80.9		48.2	
CRP	<0.5	Mean	1.2	a	1.0	a	0.9	ab	0.5	b	0.5	b	0.5	b
mg/L		sd	0.8		0.5		0.6		0.1		0.2		0.1	

Abbreviations: ALT, alanine aminotransferase; ALP, alkaline phosphatase; GGT, gamma-glutamyl transferase; T0, before administration of supplement; T1, T2, T3, T4, and T5 correspond to 7, 14, 21, 28, and 35 days from the administration of supplement; a, b on the same column denote statistical differences for *p* < 0.05. ns: not significant.

**Table 4 animals-14-00579-t004:** Sum of short-chain fatty acid, and lactic acid (Total) and molar partitioning (%) in the feces of dogs with liver disease before and after the administration of supplement.

Time	Total μmol/g	Lactic %	Acetic %	Propionic %	Isobutyric %	Butyric %	Isovaleric %
T0	213.9 ^ab^	2.1 ^ab^	66.7 ^ns^	15.9 ^ns^	3.3 ^ns^	3.5 ^ns^	8.5 ^ns^
T1	173.5 ^b^	0.4 ^b^	66.9 ^ns^	20.9 ^ns^	3.9 ^ns^	2.5 ^ns^	5.4 ^ns^
T2	227.1 ^ab^	1.4 ^ab^	69.6 ^ns^	16.4 ^ns^	4.0 ^ns^	4.5 ^ns^	4.1 ^ns^
T3	249.7 ^a^	2.6 ^a^	72.1 ^ns^	14.0 ^ns^	3.8 ^ns^	4.2 ^ns^	3.4 ^ns^
T4	225.1 ^ab^	1.9 ^ab^	66.6 ^ns^	17.4 ^ns^	4.6 ^ns^	3.2 ^ns^	6.3 ^ns^
T5	210.0 ^ab^	2.5 ^a^	67.4 ^ns^	16.7 ^ns^	4.3 ^ns^	4.1 ^ns^	5.0 ^ns^
RMSE	8.9	0.3	1.8	1.3	0.4	0.3	0.9

T0, before administration of supplement; T1, T2, T3, T4, and T5 correspond to 7, 14, 21, 28, and 35 days from the administration of supplement; a, b on the same column denotes statistical differences for *p* < 0.05. ns: not significant.

## Data Availability

Raw sequence data are available at NCBI Sequence Read Archive (PRJNA1051361).

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
