# Peer review of "Supplementation with Silybum marianum Extract, Synbiotics, Omega-3 Fatty Acids, Vitamins, and Minerals: Impact on Biochemical Markers and Fecal Microbiome in Overweight Dogs"

_animals, 2024, doi:10.3390/ani14040579_

Round 1

Reviewer 1 Report

Comments and Suggestions for Authors

General comments

Title – please include in the title what was the supplement provided to dogs

Introduction – you need to add how the other compounds in the supplement might affect liver disease in dogs.

Materials and Methods – you need to better describe some of the methods used. I have highlighted some examples on the specific comments. Why you didn’t test for changes in B. subtilis, but tested other bacteria? Why did you target these specific bacteria in your analysis?

Results – you need to be consistent in how you report the effects of the different time points you have. You cannot just discuss the ones that you have some statistical differences. For example, L214-215, you reported that there was a decrease in AST from T0 to T2; however, you didn’t mention that the values increased over time and were not different from T0 or T2. This is important to mention to not mislead the readers. If you don’t have statistical differences you should not report numerical differences, please revise that throughout the manuscript.

Discussion – there are a lot of extrapolations about what liver diseases the dogs had and how the parameters evaluated influenced these diseases. Please, review this section and remove all the extrapolations made regarding liver disease and be true to the data that you have. If the averages for the time points were not statistically different, then you cannot associate numerical differences with effects in diseases that were not diagnosed. You need to provide in the discussion what are the effects of the other nutrients added in the supplement and the parameters that you tested. What are the effects of vitamins and minerals in liver disease? Please revise your discussion to include information about these nutrients.

Specific comments

L13-15 – the 35 days was because the dogs had liver disease for 35 days or the supplement was provided for 35 days? Please review this sentence to make this clear.

L24-27 – similar as before, please be clear about what this 35-day period refers to

L35-36 I don’t think that the results underscore the usefulness of the supplement, as much as the supplement doesn’t have the effect that the authors hypothesized. The other take that I could have with this sentence, is if the methods you chose to evaluate this supplement were not appropriate, then the results would not provide the necessary evidence to properly evaluate the supplement. What was the case here?

L62-64 please add several references about functional compounds from herbs, nutraceutical plants, enzymes, and metabolites that prevent diseases.

2.1 Animals and Study Design – were all the dogs consuming the same diet before the start of the trial? Please provide the nutrient composition of the diet provided to the dogs during the trial. Because you measured the gut microbiome, diet is known to affect the gut microbiome and diet effects must be considered in this trial and that could help you to explain the variability in your microbiome data.

Table 1 – most of the dogs in your trial are overweight, with 4 out of 10 dogs being borderline obese. Obesity is a disease known to cause chronic low inflammation. How was this considered in your research? Dog weight could also have an effect on your results.

L126-127 – you mentioned a 7-day adaptation period, what was this period for?

L128-129 – most of the dogs didn’t have weights as multiple of 15kg. Please, add on table 1 how many tablets of the supplement each dog received. Did you make any correction factors for the possible differences in supplement dose each dog received?

L154-155 – please describe what were the instructions given to the owners. You can add them as supplementary material. How were the researchers sure that the owners followed the instructions? Because you analyzed for SCFA, there is a need to be extra careful on how the samples were handled because these compounds are volatile.

L158-171 – please describe the conditions of the chromatography analysis or provide a reference for the method used.

L214-215 – please add that there was as increase in AST after T2 and the results were not different from T0 or T2 after the increase.

L215-217 – if there were no statistical differences for ALT, you should not mention numerical differences, because they have no influence on your results. Please revise.

L217-218 – please add that even though there was a decrease ALP, all the average values for all time points were above the maximum for the reference range.

L218-219 – if the averages were not statistically different, there was no slight decrease observed. It is misleading reporting the data this way. Please revise

L221-222 – triglycerides did not show a similar trend as glucose because there were no statistical differences. Please revise.

L224-226 – bilirubin concentration at T2 was 0.4 as per table 3, please revise. Also, the concentration of bilirubin was within the normal range from T1 until T5, please revise.

L232-233 – please remove “BUN, blood urea nitrogen” from the list of abbreviations

Figure 1 – please increase the font size of the axis and the legend in this figure. I can’t find the black line for the “Select All” indicated in the legend, please add this line to both charts in figure 1.

Figure 2 and Figure 3 – same as figure 1, please increase the font size in the chart axis

L302 – What box-plots? Please revise

L306 – “RAs” not defined previously, please make sure all abbreviations are described when they first appear in the manuscript.

Figure 4 – T5 is missing the superscript for the statistical analysis, please revise. Increase the font size on the charts to the same font size you used in the manuscript.

L318-319 please remove this sentence from your manuscript. You cannot remove the variability of the data.

L328-329 – there was no statistical increase from T0 to T2, please make sure to properly address the numbers statistically and not numerically. Please revise this sentence.

Table 4. Please report all the values of SCFA and lactic acid in mmol/g of feces instead of mmol/L and percentage. Reporting the numbers in mmol/L of what? Please always report the values referring to the original sample. You didn’t use a volume of feces for this analysis, you used 1 gram of feces.

L337 – please report the numbers in mmol/g of feces instead of mmol/L.

L340-341 – this lowest concentration was numerical, not necessarily statistical. Please revise

L342-343 – again, this is a numerical decrease, not supported statistically, please revise this sentence.

L351 – this study did not evaluate the impact of diet on liver function. This study has no information about the diets that the dogs were eating prior to and during the study. You evaluated the effects of the supplement on some parameters that indicate liver disease, please revise the sentence

L357 – blood glucose levels were within the normal range for dogs, please revise

L358-359 please remove this sentence, you have no data to support this statement.

L362-364 you didn’t add any reference values for CRP on table 3. Thus, what is a high concentration? You need to define that and add a reference value on table 3 for comparison. Please revise this sentence

L364-366 if there were no statistical differences in triglycerides level, how this could be related to a liver disease that was not diagnosed in these dogs? Please review this sentence and be true to the data that you have from the clinical evaluation of the dogs prior to the trial and during the trial. If variations between animals of different breeds should not be the case here because you used the animal as its own control by measuring the effects over time instead of assigning the dogs to different treatment groups. Please provide more details about why the bacteria population changed according to the results you provided.

L369-370 – you can’t state that there was a positive effect on the liver just by measuring these enzymes on blood. AST values did not improve from T0 to T5. Please revise

L374-376 – in what species these beneficial effects were reported? Please revise

L376-378 – results from the literature provided are not conclusive on the benefits of this plant as a supplement for liver disease in dogs. Please revise.

L390-401 How the concentration of B. subtilis and FOS compares with the other studies cited? What was the levels of soluble fibers of the diet fed to the dogs during the trial. If the diet already had high levels of soluble dietary fiber, it is possible that the extra fiber from the supplement would not have any effect because it would not be a contribution to the total amount of fiber consumed to promote any changes.

L409-413 The variability in fecal microbiome could be attributed to the supplement in some cases, specially if the diet of the dogs did not change throughout the duration of the trial.

L430-431 only some of the blood parameters were affected by the supplement and you didn’t provide any clinical information that the disease of the dogs had signs of improvement. Please revise.

Reviewer 2 Report

Comments and Suggestions for Authors

Dear Authors,

the paper is well-presented and clear. However there are few points that should be clarified (and possibly reported as limitations of the study).

Absence of a control group.

Other  concomitant diseases were excluded?

The absence of  histological examinations of the liver represent a diagnostic limit, please clarify this aspect.

Have dogs also received a specific diet or their feeding (except for dietary restriction) remained unchanged? In the case of a specific "liver" diet, without a control group, how  do the authors attribute the improvement to the exclusive use of the supplement?

Lines 149-150: please correct the informations about the diagnostic methods (as in paragraph 2.2.2) 

Reviewer 3 Report

Comments and Suggestions for Authors

The paper by Balouei and coauthors evaluated the effect of a dietary supplement on selected blood parameters and the microbiome composition of dogs affected by liver diseases.

I have acknowledged the utility of this paper, as most dogs with liver disease are regularly fed at least one dietary supplement, in combination with standard therapy, even though the evidence of the efficacy of these supplements is still not demonstrated by rigorous studies.

However, some points need to be clarified by the authors and detailed further in the text, and clear limitations should be stated.

Please see the detailed list:

o   The study population should be clarified, particularly regarding the liver diseases diagnosed in those dogs. In the text, it’s defined as metabolic liver disease. Still, according to the most recent ACVIM consensus (DOI: 10.1111/jvim.15467), metabolic processes are two well-defined liver diseases that cannot match with those cases. The authors should clarify which kind of liver diseases they looked for in this study, and state that the absence of a liver biopsy is an evident limit, as most liver diseases can be diagnosed properly by biopsy only. 

o   According to Table 1, all but one dog were obese (and one was overweight): the authors did not comment on this, nor explain if this disease might have affected the results of the study, and how. Moreover, could the authors explain how they exclude the effect of obesity on blood parameters they have considered as a marker of liver disease, as some of them are well known to be affected by obesity? Full blood test results could be made available as supplementary files.

o   The dogs continued to be fed their regular diet. This is a limitation of the study, as diet is one of the main factors that affect the gut microbiota composition. Did the authors evaluate the confounding effects exerted by some nutraceuticals contained in the supplement as well as in the different diets fed to dogs? Most of the nutrients are regularly present in the composition of maintenance diets for dogs as well as in veterinary therapeutic diets, such as those formulated for obese dogs which might be fed to those animals (e.g. FOS, b-glucans, fish oil…). This plus the amount in the supplement led to those dogs receiving different amounts of nutrients, and it should be counted, as it could be one of the reasons why the dogs displayed such high variations between subjects in terms of microbiota composition.

o   Specify the amount of EPA and DHA included in the supplement, instead of the total amount of Omega3 in Table 2

o   Could the authors clarify why they decided to perform blood sampling every week, instead of assessing the effect of the nutritional supplement, e.g. monthly, but for a longer period than 35 days?

o   State the abbreviation RA the first time you used it.

o   Add the RI for CRP in Table 3 and consider using median [range] values, instead of RMSE, for blood variables, as more useful for evaluating the blood parameters over time points.

o   Figure 2: please resize in order to increase the readability of the graphs.

o   Line 337: please correct the sentence, as total SCFA was expressed as mmol/l, while lactic acid was in percentage.

o   The authors are invited to modify and improve the discussion according to the previous comments

Reviewer 4 Report

Comments and Suggestions for Authors

General comment: 
The bigger limit of the study is represented by the low number of cases. When we talk about microbiota, 10 cases could be inadequate and does not permit the conclusive observation. The authors tents to discuss the results in a way that may result speculative, based on this fact. 
The greatest criticality is represented by the materials and methods and the inclusion criteria which are unclear and above all in my opinion potentially incorrect with respect to the objective of the work. The results are expressed quite clearly while the discussion is vague and focuses little on the objective results of the work, without commenting on the limits. The compound is a multi-compound so it would be important, in the discussion, not to refer to a single molecule as potentially responsible for the changes found between the various timepoints.

ABSTRACT

16-17: Why should the initial reduction in diversity necessarily indicate an action on the liver-gut axis? Alpha diversity is a positive parameter associated with microbiota polymorphism.

24: “hepatic gut microbiota axis” it would be better to use another more well-known term, such as liver-gut (or gut-liver) axis and always use that.

30: Regarding glucose, the authors do not comment on this reduction in relation to a possible pathogenetic mechanism. Furthermore, the changes are clinically insignificant.

35: I would say it is better to talk about potential usefulness rather than usefulness

INTRODUCTION

General comment: I understand that they have to explain the rationale for the various things they did, but based on the way the introduction is written, I have two considerations:

- it is not clear what is known for humans and what is known in veterinary medicine, it is necessary to specify since in MV there is practically a total absence of literature and this is absolutely not evident from how the sentences are constructed

- the feeling is that it is a bit of a patchwork and that there is a lack of connection between the various paragraphs, I don't know how and if this can be improved, but a lot of the information seems disconnected to me. There is a lack of logical succession of the various concepts or scientific assumptions.

42-44: I would modify the sentence, it seems to me to be a strong statement since the mechanisms for the development of these pathologies are multifactorial. It would be better to refer to this multifactorial nature using relevant bibliographical references.

46-49: Here it is necessary to explain that this is known for human hepatology but that to date there are no studies with clinical trials with vitamin E and C in dogs with liver disease (they could however cite the ACVIM guidelines for CH management, which however clearly express the absence of literature).

53-55: This consideration is not relevant and pertinent for the aim of the study.

59-65: This paragraph should be stated at the beginning of introduction.

68-69: I don't agree with this statement, there are very few studies. Ex. Habermaass, V.; Olivero, D.; Gori, E.; Mariti, C.; Longhi, E.; Marchetti, V. Intestinal Microbiome in Dogs with Chronic Hepatobiliary Disease: Can We Talk about the Gut–Liver Axis? Animals 2023, 13, 3174. https://doi.org/10.3390/ ani13203174

71: Gut microbiota-liver relationship should be better called “gut-liver axis” 

82-83: It should be clearly stated that this consideration refers to human medicine. 

86-87: Which should be the prebiotic-like functions of SCFA? Can you cite some references?

97: promoting "liver function", I don't totally agree with this expression which implies liver function, I would call it more reduction of signs of damage.

And then, what would chronic METABOLIC HEPATITIS be? it's an unclear pathological entity. 

98-99: what do you mean as “nutritional and clinical parameters”, did you consider also clinical signs between the timepoints?

METHODS

General comments: INCLUSION criteria are the major critical issue. Furthermore, they do not specify what the dogs were being treated with, nor whether they were on monotherapy with this supplement. They do not even refer to the pre-study diet.

103-105: “All dogs were diagnosed with chronic metabolic hepatitis following clinical and laboratory examinations conducted by the same veterinarian.”

This sentence is not enough to say what the inclusion criteria are:

- do not refer to any bibliography

- I have no idea what chronic metabolic hepatitis is

- they must specify what the clinical, clinicopathological, ULTRASOUND, CYTO/HISTOLOGICAL parameters are for having included these dogs and having "assigned" that diagnosis to them

107: What do you mean as “pathological conditions referring to chronic metabolic hepatitis?”, are they considered chronic on which base? Clinical, hystopatological, ultrasonographic, biochemical? You suppose the presence of a lipidosis in relation to the overweight?

116-118: maintenance diet adult or senior? At the moment of inclusion, diet was different among dogs?

120-121: What do you mean as overall health and “any changes”? What did you precisely evaluated and monitor between various timepoints? Were the dogs recieving other therapies?

125-129: it is not very clear whether they change the power supply or not, because in that case it would be a great bias.

131: In my opinion, the fact that they used a multi-supplement is not excellent: it is not clear what can have more weight in determining the positive changes they describe.

137: Table 2 reports the %, but does it refers to dry matter?

140: Define “potential changes”

145-147: Define what did you consider ad sign of liver function or liver injury

149: Idexx Italy? Maybe you shold specify the Registration®

156-157: they don't talk about how the shipment took place (e.g. cold chain). For how long were the samples preserved? You did not used -80°C storage?

172-173: Is this technique validated?

214-216: it is not clear what the data reported for AST and ALT refers to (medians? Averages? The deviation indices are also missing). Are the clinical follow-up results avalible?

219-220: glucose could also decrease due to the onset of insufficiency, but I imagine that for the authors it reflects a glucose metabolic improvement? You should explain the pathogenesis of the variations that you consider to support your hypothesis.

223: Was serum cholesterol evaluated?

230: It seems to me that all the SDs and also the ranges that they assume as normal are missing

254: The low number of cases makes the evaluation of inter-individual variability scarcely interpretable and this should be clearly stated as a limit. 

309: In the Figure 4 the letter a/b is missing in the T5 timepoint of Ruminococcus gnavus

332: There is a typo (diseas), but however the definition “liver disease” is really generical and maybe not appropriate for the population included

DISCUSSION:

General comment: I don't really like how it's structured. Especially from the microbiota point of view, they have contrasting results with the human literature and which suggest that the integration has caused negative changes, however given the biases mentioned above it is difficult to hypothesize what these changes could be traced back to. It seems to me that the discussion makes many references of a more introductory nature without really focusing on the results.

The limitations of the study are not reported, while considering all that the work has, I would say that they are necessary in order to interpret the data correctly.

351-352: I still don't have a clear idea of ​​what they wanted to evaluate, whether they changed their diet or not, or whether they did part maintenance diet + part integrated diet. Then they talk about overall wellbeing but they don't remember the clinic either at t0 or at the other timepoints.

353: The term “microbiota” should be used instead of “microbiome”.

354: I noticed that ALT among your population was borderline, is it in contrast with your inclusion criteria?

355-366: I would like to see a ref. where they see this definition, I honestly don't know it. Are hyperglycemia and hypertriglyceridemia sufficient to define this kind of specific diagnosis? Is it possible to have a reference to support this?

358-359: Excessive. It's fine for them to comment on the absence of a biopsy, but they can't rush into hypothesizing the diagnoses (especially considering that they don't even remember ultrasound, cytological findings, etc...).

360-361: THIS SEEMS REALLY TOO MUCH TO ME, you can't say that the parameters reflect the histology when you haven't evaluated it!

364-366: I don't agree, if they changed the diet and/or the ration as they seem to say in M&M, the variation in these parameters (glucose, triglycerides) could be due to this or to weight loss (they don't report differences in weight and BCS between the various T).

387-388: “If the blood parameters indicted a recovery of the dogs from the liver inflammatory and metabolic chronic disease” This seems like too strong a phrase to me, I would refer to the improvement of the parameters taken into consideration, if at all. Putting it this way in my opinion is not sufficiently supported by their results and is at the limits of speculation.

405-407: You could not trace back the positive effect to Silybum marianum, considering that your nutraceutical compound contains a lot of substances and you also changed the diet.

409-411: this represent a limit and should be stated.

CONCLUSION
It seems adequate to me compared to their results (I don't like the one in the abstract, however, which should be done more along the lines of this one which is more moderate and consistent with the results)

Round 2

Reviewer 3 Report

Comments and Suggestions for Authors

After the revisions are made, there remains one additional point that needs to be clarified:

The authors stated that the dogs’ original diets remained unchanged, and only an adjustment for the amounts of food in relation to the energy requirements was done. What does it mean? Theoretically, the energy requirement of obese dogs consists of an energy intake lower than what they are used to, with the aim (or resulting in) weight loss. Considering that the authors originally looked at the effect of the supplement in relation to a potential liver disease and not to obesity, weight loss was probably not considered. Did the authors weigh the dogs throughout the study to assess any changes in body weight? Basically, the authors should clarify here and in the text what they intended with “adjustments of food amount”. It’s an important point, as weight loss should be the first intervention to manage obese dogs, and it can only be pursued by applying a caloric restriction. Feeding supplements, even though with anti-inflammatory or anti-oxidant properties, as a first-line treatment for obese dogs, rather than caloric restriction, has a very low relevance from a clinical point of view.

I suggest moving the limitation part from the conclusion to the discussion.

Reviewer 4 Report

Comments and Suggestions for Authors

Round 3

Reviewer 3 Report

Comments and Suggestions for Authors

The authors were able to make the revisions that the study design would admit. 

Comments on the Quality of English Language

english is fine

Author Response

Thank you for your collaboration.